



# Determining Cloud Thermodynamic Phase from the Polarized Micro Pulse Lidar

Jasper R. Lewis[1,2], James R. Campbell[3], Simone Lolli[4], Sebastian A. Stewart[5], Ivy Tan[1,2], Ellsworth J.
Welton[2]

[1]Joint Center for Earth Systems Technology, University of Maryland Baltimore County, Baltimore, Maryland, USA
[2]NASA Goddard Space Flight Center, Greenbelt, Maryland, USA
[3]Naval Research Laboratory, Monterey, California, USA
[4]CNR-IMAA, Istituto di Metodologie per l'Analisi Ambientale, Tito Scalo, Italy
[5]Science Systems and Applications, Inc., Lanham, Maryland, USA

*Correspondence to*: Jasper R. Lewis (jasper.r.lewis@nasa.gov)

**Abstract.** A method to distinguish cloud thermodynamic phase from polarized Micro Pulse Lidar (MPL) measurements is
described. The method employs a simple enumerative approach to classify cloud layers as either liquid water, ice water, or
mixed-phase clouds based on the linear volume depolarization ratio and cloud top temperatures derived from Goddard Earth
Observing System, version 5 (GEOS-5) assimilated data. Two years of cloud retrievals from the Micro Pulse Lidar Network
(MPLNET) site in Greenbelt, MD are used to evaluate the performance of the algorithm. The fraction of supercooled liquid
water in the mixed-phase temperature regime (-37 °C – 0 °C) calculated using MPLNET data is compared to similar
calculations made using the spaceborne Cloud-Aerosol Lidar with Orthogonal Polarization (CALIOP) instrument on board
the Cloud-Aerosol Lidar and Infrared Pathfinder Satellite Observations (CALIPSO) satellite, with reasonable consistency.

## 20   1 Introduction

Due to their high temporal and vertical resolutions, and unique spectral sensitivity, lidars are key instruments for
atmospheric profiling of gaseous species, aerosols and translucent clouds. In addition to providing unambiguous layer height
information, lidars are used for retrievals or direct measurements of backscatter, extinction, optical depth, temperature, and
concentrations of these respective atmospheric constituents (Weitkamp, 2005). Polarized lidar systems transmit light in one
linear state and by use of an optical device, typically a beam splitter, detect the returned signal from both the initial and
orthogonal polarization states. The ratio of these two signals is referred to as the linear depolarization ratio (LDR),

$$\delta = \frac{P_\perp}{P_\parallel}, \tag{1}$$

where $P_\perp$ is the signal measured from the orthogonal polarized state and $P_\parallel$ is that from the signal parallel to the initial
polarization state. From the time the earliest polarized lidar measurements were made, it was realized that the LDR could be
used to distinguish certain atmospheric constituents (Cohen et al., 1969; Schotland et al., 1971; Pal and Carswell, 1973).



Specifically pertaining to clouds, liquid water clouds exhibit low LDRs (near zero) because of their spherical shape; while ice water clouds, due to their irregular shape, tend to have higher values (between 0.3 – 0.6) and mixed-phase clouds exhibit LDRs in-between these two extremes (Sassen, 2005 and references therein). It is noted that multiple scattering induces an increase in the apparent LDR measured increasingly further into the clouds (Sassen and Petrilla, 1986; Sassen, 1991; Hu et al., 2006), which can lead to values for liquid water clouds approaching the threshold for ice water clouds with increasing depth. Conversely, oriented ice plates produce relatively low LDRs that can be mistaken for liquid water clouds if the lidar is not tilted slightly off-zenith (Sassen, 1991).

Reliable, long-term observations of cloud thermodynamic phase are critical for studies of the Earth's radiation budget. Liquid water clouds are broadly characterized by relatively warmer temperatures, smaller droplet sizes and higher number concentrations. Therefore, they are more efficient at reflecting shortwave radiation and are generally associated with an overall negative cloud radiative effect (CRE) or cooling (Yi et al., 2017). Conversely, ice water clouds (and specifically cirrus clouds) are broadly characterized by colder temperatures, larger particle sizes, and lower number concentrations. Therefore, they can be more efficient at trapping longwave radiation and are generally associated with an overall positive CRE or warming, though its magnitude and sign exhibit latitudinal and daytime temporal diurnal variations (Campbell et al., 2016; Lolli et al., 2017; Campbell et al., 2020). The CRE of mixed-phase clouds will vary depending on the ratio of ice to liquid within the cloud (Sun and Shine, 1994; Korolev et al., 2017).

Future changes in Earth's climate may result in changes in the occurrence and global distribution of cloud types (Stephens, 2005; Hu et al., 2010; IPCC, 2013), so it is important to record and monitor cloud phases across all climate regions. Furthermore, current numerical weather prediction and climate models misrepresent cloud phase (particularly, ice and mixed-phase) as seen by observations because the processes that govern phase transitions are still not fully understood (Ramanathan et al., 1989; Ringer et al., 2006; IPCC, 2013; Tan et al., 2016; Costa et al., 2017). Because these processes take place on spatial scales much smaller than model grid sizes, more frequent and diverse observations are needed to improve cloud parameterizations.

The National Aeronautics and Space Administration (NASA) Micro Pulse Lidar Network (MPLNET) is a federated network of Micro Pulse Lidar (MPL) systems deployed worldwide in support of basic science and the NASA Earth Observing Systems (EOS) program (Wielicki et al., 1995; Welton et al., 2001). Since beginning in 2000, MPLNET has operated using a standardized instrument and common suite of data processing algorithms with thorough uncertainty characterization, which makes for straightforward comparisons between sites. Most MPLNET sites are collocated with the Aerosol Robotic Network (AERONET), providing profile and column measurements of aerosols and clouds in tropical, mid-latitude, and polar climate regions (Holben et al., 1998; Welton et al., 2002; Campbell et al., 2003). Following the modified EOS convention, data are

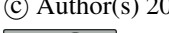

publicly available at Level 1 (L1; near real time, no quality screening), Level 1.5 (L15; near real time, quality screened) and
Level 2 (L2; upon request, not real time) product levels (http://mplnet.gsfc.nasa.gov).

The lidar signal data, normalized relative backscatter (NRB; Campbell et al., 2002; Welton and Campbell, 2002), are utilized
in the processing of all other MPLNET products (i.e. aerosols, clouds, planetary boundary layer). The Version 3 (V3)
MPLNET cloud algorithm is described fully by Lewis et al. (2016). Cloud layer height retrievals are performed using two
methods. The first relies on gradients in the lidar backscatter profile and is primarily used for low-level liquid water phase
clouds. The other uses the uncertainty in the lidar signal, as described by Campbell and Sassen (2008), and is primarily used
for high-level clouds (i.e. cirrus). A multi-temporal averaging scheme is used to improve high-altitude cloud detection
beyond the previous Version 2 cloud algorithm. In addition to layer height information, the V3 cloud products include
estimates of extinction and optical depth for thin cirrus clouds, cloud fractions and cloud thermodynamic phase. Polarized
MPLs were introduced to the network at the time Lewis et al. (2016) was written; however, the depolarization variables were
still in development and not used as part of the algorithm. The goal here is to present a method by which ice water, liquid
water, and mixed-phase clouds can be identified from polarized MPL measurements to fully describe the cloud
thermodynamic phase.

## 2 Determining cloud thermodynamic phase

### 2.1 Polarized micropulse lidar data

The concept of a polarized MPL was introduced by Flynn et al. (2007). The original design used a single detector and a
nematic liquid crystal retarder (LCR) to switch between linearly and circularly polarized states. However, the LCR was
limited to millisecond switching speeds, at best, which was too slow for some cloud observations, and generally unwieldy
overall relative to the data acquisition system available for MPL instruments at the time. Therefore, the original polarized
MPL design was never used within MPLNET. A new design, using a ferroelectric liquid crystal (FLC) to provide switching
speeds on the order of microseconds, has been thoroughly tested and characterized within MPLNET and is the basis for the
new cloud thermodynamic phase algorithm. Polarized MPL data have previously been used to autonomously detect light
precipitation (Lolli et al., 2013, 2020). The polarized MPL requires temperature and polarization calibrations to reduce
systematic biases in the measured signal and depolarization ratio to within fractions of a percent. Without proper calibration,
systematic biases as large as 30% may occur (Welton et al., 2018).

Despite the design change, the data produced using the FLC is similar to that shown by Flynn et al. (2007), and the
relationships given to obtain the total lidar signal power and LDR are still applicable. Here, and from this point forward, we
refer exclusively to the linear volume depolarization ratio, which includes contributions from both particulate and molecular



backscatter. This is in contrast with retrievals of the linear particle depolarization ratio, which removes the molecular

contributions. The total lidar signal is given by the normalized relative backscatter (NRB),

$$\text{NRB}(z) = P_{co}(z) + 2P_{cross}(z), \tag{2}$$

where $P_{co}$ is the co-planar signal and $P_{cross}$ is the cross-planar signal. The LDR, equivalent to (1), and its uncertainty are

given by

$$\delta(z) = \frac{P_{cross}(z)}{P_{co}(z) + P_{cross}(z)} \qquad \text{and} \tag{3}$$

$$\Delta\delta(z) = \sqrt{\delta(z)^2 \left[ \left( \frac{\Delta P_{cross}(z)}{P_{cross}(z)} \right)^2 + \left( \frac{\Delta P_{co}^2(z) + \Delta P_{cross}^2(z)}{(P_{co(z)} + P_{cross}(z))^2} \right) \right]}. \tag{4}$$

An example of the NRB and LDR measurements collected at the NASA Goddard Space Flight Center (GSFC) site on 5 Oct

2019 are shown in Fig. 1. The LDR in Fig. 1 suggests the presence of supercooled liquid water/mixed-phase stratified

clouds around 9 km, liquid water clouds below 2 km, and ice water (cirrus) clouds near 11 km toward the end of the day.

**2.2 Algorithm description**

Cloud thermodynamic phase is determined using the LDR and its uncertainty, and the cloud top temperature (CTT) obtained

from the Goddard Earth Observing System, version 5 (GEOS-5), atmospheric general circulation model (AGCM; Rienecker

et al. 2008; Molod et al. 2012). Specifically, the Forward Processing for Instrument Teams (FP-IT) GEOS-5, version 5.9.1,

data are utilized (http://gmao.gsfc.nasa.gov/products). A schematic of the cloud phase algorithm is shown in Fig. 2. The

first step in the process is to obtain the LDR and uncertainty for each altitude bin within the detected cloud layer. The reason

for using individual altitude bin values instead of layer-integrated values is to avoid the ambiguity that exists for mixed-

phase clouds due to the stronger signal return from liquid water compared to that from ice crystals. An example of this is

shown in Fig. 3 for the mixed-phase cloud presented from 5 Oct 2019 at GSFC. The cloud layer observed between 8 − 9 km

(CTT = -32.1 °C) exhibits NRB near the cloud top in both the co-polar and cross-polar signals compared to the signals nearer

the cloud base. However, the co-polar signal peaks to almost two orders of magnitude larger than the cross-polar signal at

the cloud top. The resulting LDR is nearly 0.3 just above the cloud base (indicative of precipitating ice crystals) and less

than 0.02 at the cloud top (indicative of liquid water). In contrast, the layer-integrated LDR,

$$\bar{\delta} = \frac{\int_{base}^{top} P_{cross}(z)\, dz}{\int_{base}^{top} [P_{co}(z) + P_{cross}(z)]\, dz}, \tag{5}$$

has a value of 0.035 that could be mistakenly identified as pure liquid water cloud phase. The ability to detect mixed-phase

clouds in this manner is unique to ground-based lidar systems. Spaceborne lidar (e.g. Cloud-Aerosol Lidar with Orthogonal

Polarization or CALIOP) view clouds like the one shown in Fig. 3 from above and thus risk the signal being attenuated

within the liquid water portion of the cloud, before reaching the underlying ice virga. As such, there is the potential for

CALIOP to misidentify mixed-phase clouds as consisting solely of liquid water (Zhang et al., 2010).






Figure 4 shows the relationship between LDR and temperature derived using the altitude bins resolved within each cloud layer detected using the combined V3 algorithm at GSFC from 2018 – 2019. As explained in Lewis et al. (2016), strong aerosol layers at high altitudes can be misclassified as cloud layers due to their highly variable scattering ratios. Given recent pyrocumulonimbus and volcanic activity in the stratosphere (Peterson et al., 2017, 2018; Kirin et al., 2019; Torres et al., 2020), cloud retrievals in this study are limited to the troposphere in order to reduce the impact of false cloud retrievals. The LDR in Fig. 4 is averaged in 5-C° increments and median values are plotted along with the interquartile range (IQR). The increase in LDR with decreasing temperature is qualitatively similar to Fig. 10 within Yorks et al. (2011), though they use layer-integrated values. The LDR at warmer (colder) temperatures most likely associated with liquid (ice) water clouds remains below 0.05 (above 0.30). Based on these results, each altitude bin is assigned a cloud phase diagnostic (CPD) value based on the LDR and its uncertainty as defined in Table 1. This diagnostic value provides the likely cloud phase for each altitude bin. An enumerative approach is then used to determine the thermodynamic phase of the entire cloud layer, based on the CTT and CPD.

Accurately measuring the cloud top with ground-based lidar is problematic (Pal et al. 1992; Platt et al. 1994). Optically-thin clouds can be penetrated by the laser pulse. The transition to molecular signal above the cloud may then be used to report the true cloud top. However, many optically-thicker clouds completely attenuate the lidar signal and only an apparent cloud top can be reported (Lewis et al., 2016), which produces an inherent warm bias in the CTT. Nevertheless, the cloud thermodynamic phase is presumed to be liquid water for all clouds, regardless of the CPD, if the CTT is warmer than 0 °C. Similarly, cloud phase is presumed ice water (cirrus genus) for all clouds with CTT colder than -37 °C (Sassen and Campbell, 2001; Campbell et al., 2015). Cirrus clouds are unaffected by the warm CTT bias, because only ice water is physically possible at colder temperatures. However, the presumption of liquid-water phase based on CTT alone has an unknown influence on phase retrievals of optically-attenuated clouds warmer than 0 °C (less than 5% of the GSFC sample). In such cases, ice water may very well exist above the apparent cloud top. But since the necessary information is not contained in the lidar return, supplementary data (e.g. from radar) are needed to make such a determination.

150

Only clouds in the temperature regime where water can exist in either liquid water, ice water or some combination of those use the CPD to classify the thermodynamic phase. As mentioned previously, multiple scattering effects can induce increases in the LDR of liquid water clouds to values similar to that of ice water clouds. Though the narrow field-of-view of the MPL (~100 µrad) minimizes such effects, the reliability of the CPD to detect ice is limited to a certain height above the cloud base, denoted as $\Delta h$. The value of $\Delta h$ is empirically determined as the height where the estimated two-way transmittance falls below 0.25, calculated as described by Lewis et al. (2016) using the iterative equation,

$$T_c^2(Z_k) = T_c^2(Z_{k-1})\exp\left\{-2S^*\left[\frac{R'(Z_k)}{T_c^2(Z_{k-1})}\right]\beta_m(Z_k)\Delta z\right\}, \tag{6}$$



where $S^*$ is the effective extinction-to-backscatter ratio, $R'$ is the attenuated backscatter ratio, $\beta_m$ is the molecular backscatter determined from GEOS-5, $\Delta z$ is the range resolution of the instrument, and $Z_k$ is the altitude of bin $k$ above the cloud base. At the cloud base, we assume $T_c^2(Z_0) = 1$. Retrievals from the $2018 - 2019$ GSFC data, exhibited mean values of $\Delta h$ ranging from 0.4 km (for liquid water clouds) to 1.2 km (for ice water clouds) within the mixed-phase temperature range (-37 °C – 0 °C).

The remainder of the cloud phase algorithm simply counts the occurrences of the CPD to determine a classification for the cloud layer. If multiple *ice* bins are found within $\Delta h$, then we inspect above the last *ice* bin for the presence of *liquid* or *mixed* bins (i.e. a decrease in the LDR) to determine if the layer is pure ice or mixed-phase. If no *ice* bins are found within $\Delta h$ but multiple *liquid* bins are present, then we look within $\Delta h$ for the occurrence of *mixed* bins to determine if the layer is pure liquid or mixed-phase. If neither *ice* nor *liquid* bins are found, the layer is classified as undetermined if more than 25% of the bins CPD are *undetermined* or mixed-phase, otherwise. Figure 5 shows a mask of the retrieved cloud thermodynamic phase for the 5 Oct 2019 case presented in Section 2.1. The liquid water clouds below 2 km and the ice water clouds near 11 km are classified using only the CTT. The supercooled water clouds and mixed-phase clouds are effectively classified using the enumerative approach.

## 3 Results

### 3.1 Frontal cloud example

Well-established temperature thresholds are used to classify thermodynamic phases in absolute terms for liquid water (warmer than 0 °C) and ice water (colder than -37 °C) clouds; therefore, we focus attention here on clouds occurring in the ambiguous mixed-phase temperature regime between these two temperature thresholds. Figures 1, 3, and 5 illustrate an example of stratified liquid and mixed-phase clouds in the mixed-phase temperature regime. To provide an example with very different synoptic conditions, Fig. 6 shows a frontal cloud occurring on 27 Mar 2018. Frontal cloud systems are common in the mid-latitudes and may contain any combination of liquid, ice, and mixed-phase clouds (Hogan et al., 2003; Costa et al., 2017).

The anvil cloud structure at the beginning of the day is consistent with convection and is classified as ice from the CTT, which is also consistent with high LDRs. As the cloud base descends below 7 km, the cloud phase alternates between ice and mixed-phase clouds and is classified as mostly liquid water clouds below 3 km. The limitations of using only ground-based lidar to retrieve thermodynamic phase are evident as the signal is attenuated within optically-thick liquid water and ice clouds, which results in under sampling of the atmospheric column above such clouds. Furthermore, precipitation (starting at 17 UTC) reaching near the surface is occasionally included as part of the cloud layer, which may affect the quality of the





cloud phase retrieval. For instance, raindrops have an irregular shape that enhances the LDR (Lolli et al., 2020). Therefore,
precipitation included within in a true liquid-phase cloud might be interpreted as a mixed-phase cloud.

## 3.2 Cloud thermodynamic phase statistics

Two years of GSFC cloud data (2018 – 2019) are used to examine cloud thermodynamic phase statistics derived and
prescribed from the method described in the previous section. Figure 7 shows the distribution of LDRs for each altitude bin
within cloud layers detected during the two-year period. The bimodal distribution shows two peaks at ~0.01 and ~0.37
representing the liquid water and ice water cloud phases, respectively. The fractional probability for liquid water clouds also
peaks near ~0.01 and a very small percentage of liquid water clouds contain LDRs with values above 0.1. The fractional
probability for ice water clouds has a clear minimum within the range where liquid water clouds are expected (0 – 0.05).
However, the fractional probability everywhere else typically remains above 0.50. This is partially attributed to sampling, as
there are many more bins within ice clouds than other phases because the lidar signal does not attenuate as quickly in such
layers.

Table 2 indicates the number of layers and altitude bins associated with each of the cloud phases. Another consideration is
that ice layers (especially those including virga streaks) are generally more tenuous and, because linear volume
depolarization values are used, the contribution from molecular backscatter becomes more significant. As a result, the LDR
for individual altitude bins can be much lower than what is typically expected for pure ice. Mixed-phase clouds peak at 0.05
and skew right until ~0.47. Though they represent a small percentage of the distribution, undetermined phase cases most
frequently occur with negative LDRs. While the layer-integrated LDR is not used in the algorithm, the mean values shown
in Table 2 agree well with the median LDRs for each cloud thermodynamic phase.

Figure 8 shows the distribution of CTTs and fractional probabilities of each cloud thermodynamic phase collected at GSFC
(2018 – 2019). The large majority of ice water clouds (nearly 90%) are found using the -37 °C CTT threshold only.
Similarly, but to a lesser extent, most liquid water clouds (54%) are found using only the 0 °C threshold. Within the mixed-
phase temperature regime, where water can exist as pure liquid, pure ice, or some combination of the two, liquid and ice
water distributions show an inverse relationship. As a qualitative comparison, Campbell et al. (2015; see their Fig. 1)
present similar analysis using CALIOP observations. They find that the fractional probabilities of liquid and ice water
clouds intersect near -27 °C, which is colder than the intersection point in this work (-22 °C). In addition to the different
methodologies used to determine the cloud thermodynamic phase, the instruments also have different viewing geometries
(zenith for MPLNET and nadir for CALIOP), footprints, and sensitivities that prevent any quantitative comparisons.
Coopman et al. (2020) use passive spaceborne sensors to determine the glaciation temperature at which ice and liquid equal
50% and report a global value of -24 ± 1 °C. The fractional probability for mixed-phase clouds in this work peaks near -22
°C, while undetermined phase remains relatively flat and is less than 7% at all temperatures. The shape of the mixed-phase


distribution is similar to that found by Shupe et al. (2006; see their Fig. 5) for Arctic mixed-phase clouds, though the peak shifts toward warmer temperatures in their study.

**3.3 Supercooled liquid fraction**

Much attention has been paid to the amount of supercooled liquid water in the mixed-phase temperature regime (Choi et al., 2010; Hu et al., 2010; Tan et al., 2014; Tan et al., 2016; Tan and Storelvmo, 2019; Wang et al., 2019). As liquid water presence decreases, so generally does the cloud albedo, which results in a reduced solar-reflective cooling effect. Additionally, cloud lifetime and precipitation are governed by the transition from liquid water to ice (Korolev et al., 2017). Studies have shown that low biases in the amount of supercooled liquid present in climate models leads to

misrepresentations of the outgoing shortwave radiation and feedback response to a doubling of $CO_2$ (Furtado et al., 2016; Tan et al., 2016).

The CALIOP instrument on board the Cloud-Aerosol Lidar and Infrared Pathfinder Satellite Observations (CALIPSO) satellite (Winker et al., 2010) can estimate the global distribution of supercooled liquid water in the atmosphere. Choi et al (2010) and Tan et. al. (2014) use CALIOP retrievals to examine the supercooled liquid fraction (SLF) or the ratio of the

number of liquid-phase footprints to the number of the total number of footprints (liquid-phase + ice-phase) within a specified grid box and isotherm. Similarly, we define the MPLNET SLF as the ratio of the number of liquid-phase cloud layers to the total number of cloud layers (liquid-phase + ice-phase + mixed-phase) for a specified isotherm. Because MPLNET includes mixed-phase as a possibility, without partitioning ice from liquid, the resulting SLF represents a lower-

limit on the presence of liquid water in the atmosphere. The repercussions of this distinction from the CALIOP SLF will be discussed further.

A comparison of SLFs derived from each instrument (CALIOP and MPLNET) averaged from 2015 – 2019 is shown in Fig. 9. The CALIOP SLFs were calculated for a 2.5° latitude × 5.0° longitude grid box using the procedure described by Tan et

al. (2014). The version 4.20, level 2 Vertical Feature Mask (VFM) product was used in conjunction with National Centers for Environmental Prediction (NCEP)-Department of Energy (DOE) Reanalysis 2 air temperature and pressure data (Kanamitsu et al., 2002) at a resolution of ~2.5° latitude × 2.5° longitude. Only nighttime CALIOP retrievals are used in order to avoid artifacts from solar noise. The CALIOP SLFs below -10 °C are excluded because strong lidar return-signal attenuation from clouds at these temperatures leads to significant measurement errors (Choi et al., 2010).


Comparisons between ground-based and spaceborne lidars are difficult, because the satellite moves quickly over the stationary point-source of the ground-based lidar. Satellites, like CALIPSO, provide good spatial coverage, but poor temporal sampling. In contrast, ground sites in MPLNET provide poor spatial coverage globally; but, continuous observations at 1-minute data rate provide full diurnal sampling. Low, attenuating clouds also obstruct the view of high



clouds from the surface that are easily observed from space. These two factors result in very different sampling volumes for ground-based and spaceborne measurements. Furthermore, as demonstrated by the example in Fig. 3, the opposite viewing geometries may lead to differing cloud phase classifications, even if the same cloud is observed from both platforms. Despite these unpreventable differences, Fig. 9 demonstrates that MPLNET and CALIOP (at least qualitatively) observe very similar patterns in regard to SLF. The inset of Fig. 9 also suggests that the correlation lengths for SLF may be rather

large, based on the similar values for adjacent grid boxes.

We note that the CALIOP SLFs are always higher than and outside the standard error of MPLNET SLFs at all isotherms warmer than -30 °C, but nearly match MPLNET at colder temperatures where liquid-phase is less likely to exist. A possible explanation for this difference (aside from those mentioned previously) is the potential misclassification of mixed-phase

clouds as liquid-water by CALIOP (Zhang et al., 2010). It is also plausible that MPLNET is underestimating the presence of liquid-water phase at warmer temperatures, due to precipitating clouds, as indicated in the frontal cloud example presented in Sect. 3.1. The final consideration follows from the inclusion of mixed-phase in the MPLNET SLF, that is not present in the CALIOP VFM. As stated above, the MPLNET SLF represents a lower-limit because the percentage of liquid water in the mixed-phase layer is undetermined. Therefore, it is reasonable for the MPLNET SLFs to be lower than the CALIOP SLFs at

warmer temperatures. However, more extensive analysis is necessary in order to address these differences with any certainty. Such analysis is beyond the scope of the current work, but warrants exploration in a future study.

## 4 Discussion and summary

The radiative impact of clouds is known to depend on the partitioning of liquid and ice phases (Sun and Shine, 1994; Korolev et al., 2017). However, sparse local observations have limited the amount of information necessary to evaluate and

improve model parameterizations (Matus and L'Ecuyer, 2017). Mixed-phase clouds, which occur in all climate regions and multiple cloud types, are particularly not well understood. Polarized lidar has the ability to provide vertical profiles of cloud structure, at least to the limit of signal attenuation, and add insight as to how ice and liquid water are partitioned in the atmosphere.

This work introduces a simple, enumerative method to determine the cloud thermodynamic phase from polarized Micro Pulse Lidar (MPL) measurements. In addition to the typical liquid and ice phases, we also attempt to assign mixed-phase to cloud layers within the -37 °C – 0 °C temperature regime. The zenith-viewing geometry and narrow field-of-view of the MPL make such classifications possible, though low-level liquid water clouds may inhibit observations of the full atmospheric column. Results using two years of cloud observations at the Greenbelt, MD site are at least qualitatively

consistent with previous studies of thermodynamic phase distributions. A five-year comparison with Cloud-Aerosol Lidar





with Orthogonal Polarization (CALIOP) showed reasonable agreement. However, a more extensive, long-term study involving multiple MPLNET sites is needed in order to address the differences between the complementary observations.

Though the polarized MPL was fairly new at the time an advanced cloud algorithm for MPL was introduced by Lewis et al. (2016), the instrument has since been fully tested and characterized and the Micropulse Lidar Network (MPLNET) is now fully polarized. The ability to provide continuous observations of cloud properties, including thermodynamic phase, across all climate regions using a standardized instrument and retrieval process is a distinctive feature of MPLNET. In a future work, we endeavor to explore how cloud properties differ amongst MPLNET sites. Such studies have already been performed investigating the cirrus cloud radiative effect at tropical, mid-latitude, and polar MPLNET sites (Campbell et al., 2016; Lolli et al., 2017; Campbell et al., 2020).

In closing, it must be noted that no one instrument or platform will be able to fill the void in our understanding of cloud thermodynamic phase. The results presented here have highlighted some of the strengths and limitations of ground-based and spaceborne lidar retrievals. However, it is fundamentally required to use a synergetic approach (combining in-situ and remote sensing, passive and active sensors, observations and models, etc.) in order to gain a better perspective of how liquid and ice phases are partitioned and transition from one phase to another in the atmosphere. Adding to the complexity, there is no one definition for mixed-phase clouds that can be universally applied. Instead, the definition or threshold for mixed-phase depends on the spatial and temporal resolutions and sensitivities associated with each observational method, making it even more important to use multiple, simultaneous measuring techniques to grasp the "big-picture". The cloud thermodynamic phase data presented in this work, along with the other MPLNET datasets (some sites with 10+ years of data), offer a valuable piece of the picture for long-term studies of clouds and aerosol-cloud interactions.

### Acknowledgements

The NASA Micro Pulse Lidar Network is funded by the NASA Earth Observing System and the NASA Radiation Sciences Program. The GEOS-5 meteorological data were provided by the NASA Global Modeling and Assimilation Office (GMAO) at GSFC. CALIPSO data were obtained from the NASA Langley Research Center Atmospheric Science Data Center.

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



**Table 1: Cloud phase diagnostic**

| CPD | Likely phase | Definition |
|---|---|---|
| 1 | no cloud | – |
| 2 | *liquid* | $\delta - \Delta\delta \geq 0.00$ and $\delta + \Delta\delta \leq 0.05$ |
| 4 | *ice* | $\delta - \Delta\delta \geq 0.30$ and $\delta + \Delta\delta \leq 0.50$ |
| 8 | *mixed* | $\delta - \Delta\delta > 0.05$ and $\delta + \Delta\delta < 0.30$ |
| 16 | *undetermined* | All others, including $\Delta\delta / \delta > 1.0$ |



**Table 2: Cloud phase properties for GSFC, 2018 – 2019. Number of layers ($N_{Layers}$) and altitude bins within each layer ($N_{Bins}$),**
**mean layer-integrated LDR ($\overline{\delta}_{Layers}$), median LDR ($\delta_{Bins}$) and interquartile range (IQR) for all altitude bins within each layer, and the mean cloud top temperature (CTT).**

| Phase | $N_{Layers}$ (%) | $N_{Bins}$ (%) | $\overline{\overline{\delta}}_{Layers}$ Mean ± St. Dev. | $\delta_{Bins}$ [IQR] | CTT (°C) |
|---|---|---|---|---|---|
| Liquid | 146 983 (29.0) | 996 025 (14.5) | 0.018 ± 0.017 | 0.013 [0.007, 0.024] | -0.7 ± 10.3 |
| Mixed | 58 816 (11.6) | 1 001 234 (14.6) | 0.161 ± 0.125 | 0.207 [0.074, 0.310] | -19.4 ± 9.6 |
| Ice | 294 227 (58.0) | 4 796 681 (70.0) | 0.306 ± 0.105 | 0.336 [0.248, 0.390] | -51.5 ± 12.1 |
| Undetermined | 7 293 (1.4) | 56 068 (0.8) | 0.093 ± 0.122 | 0.088 [0.008, 0.266] | -18.0 ± 11.3 |



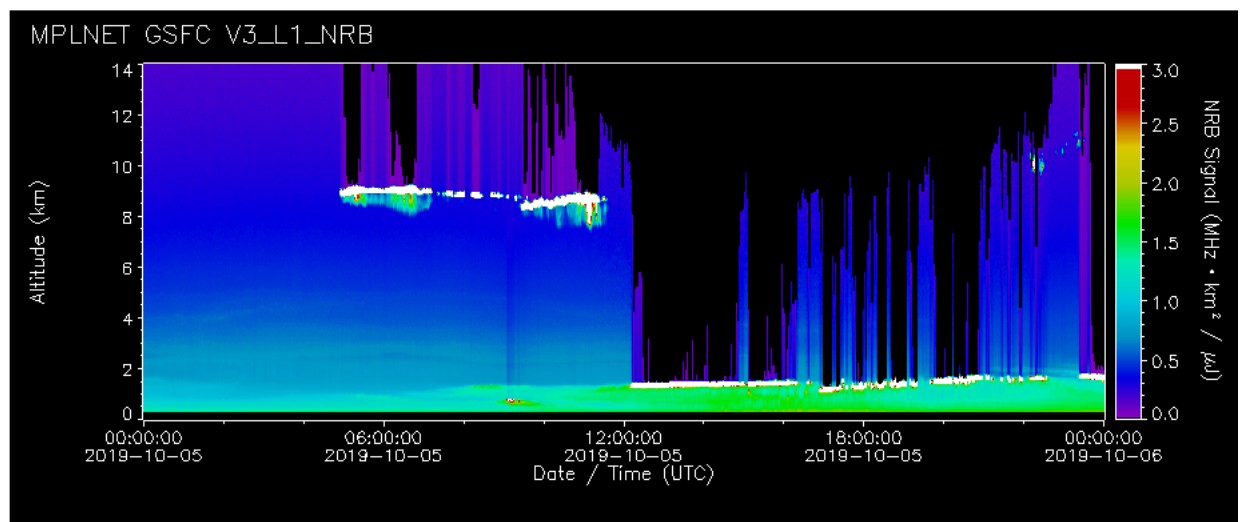

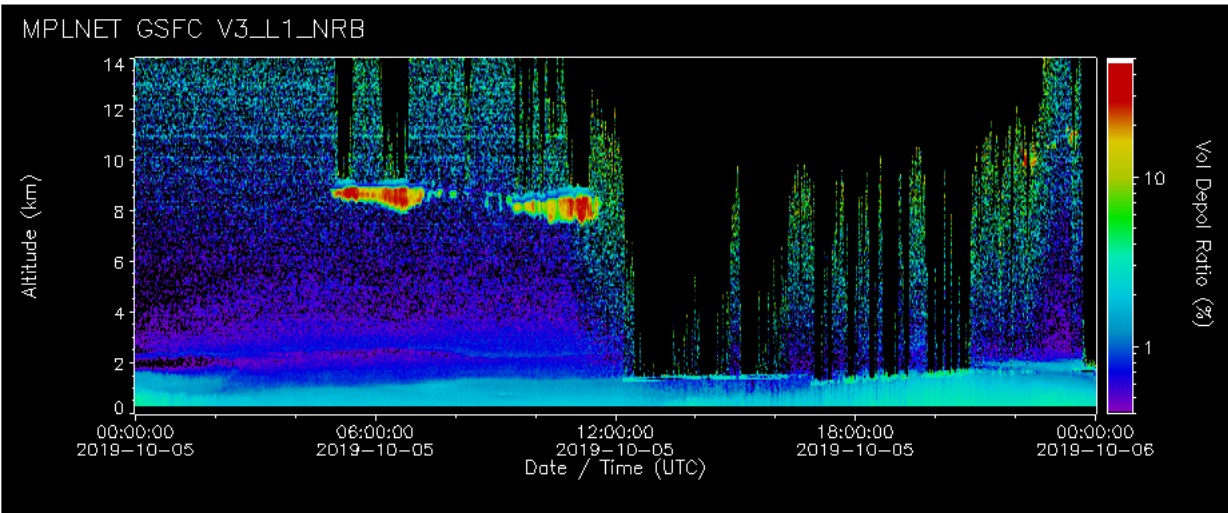


**Figure 1: Examples of the NRB (top) and volume depolarization ratio (bottom) at GSFC on 5 Oct 2019. Altitude bins where the signal uncertainty is twice the signal strength have been suppressed for easier viewing.**





**Figure 2: Schematic of the cloud phase algorithm. Lowercase phases (e.g. *ice*) indicate the cloud phase diagnostic (CPD) for individual altitude bins and capitalized phases (e.g. Ice) indicate the phase determination for the entire cloud layer.**



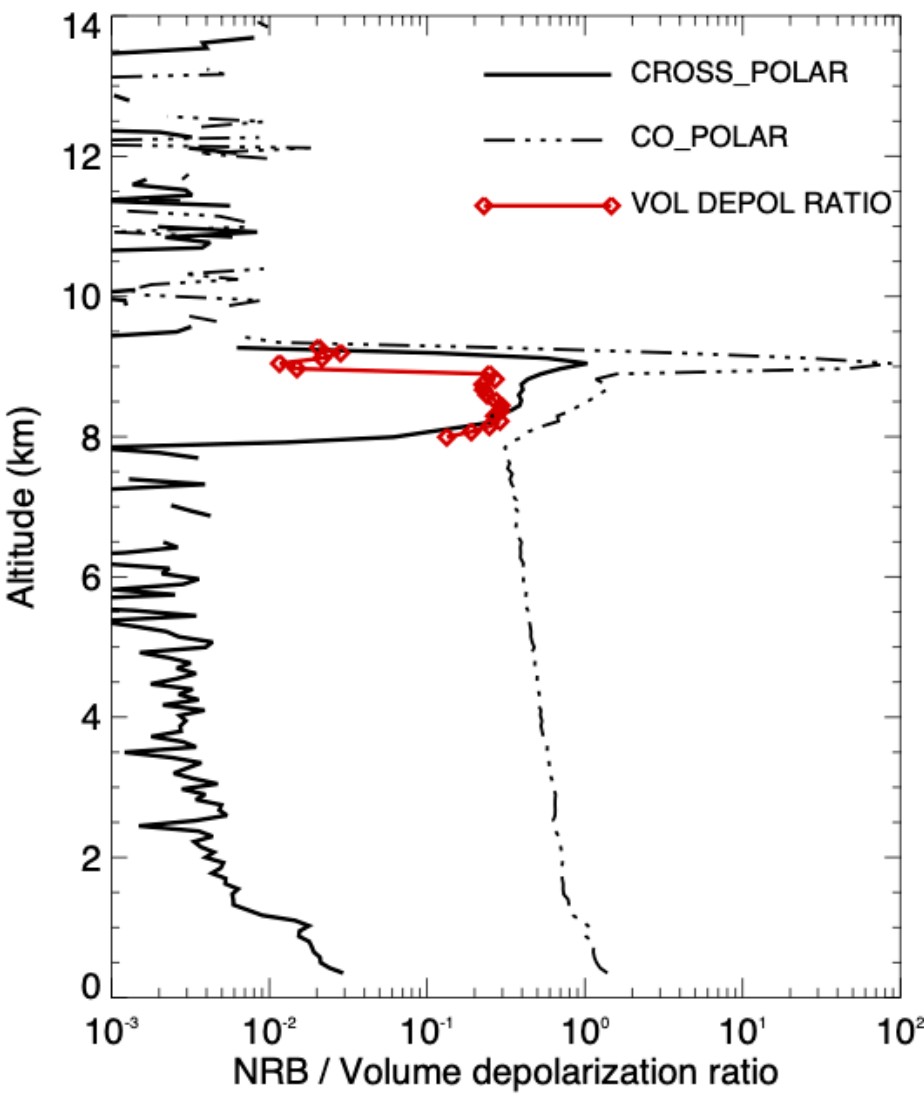

**Figure 3: Profiles of the cross-polar (solid line) and co-polar (dash-dotted line) components of the NRB for a mixed-phase cloud at GSFC on 5 Oct 2019 (6:40 UTC). The volume depolarization ratio within the cloud layer is indicated by the red line (diamond symbol).**

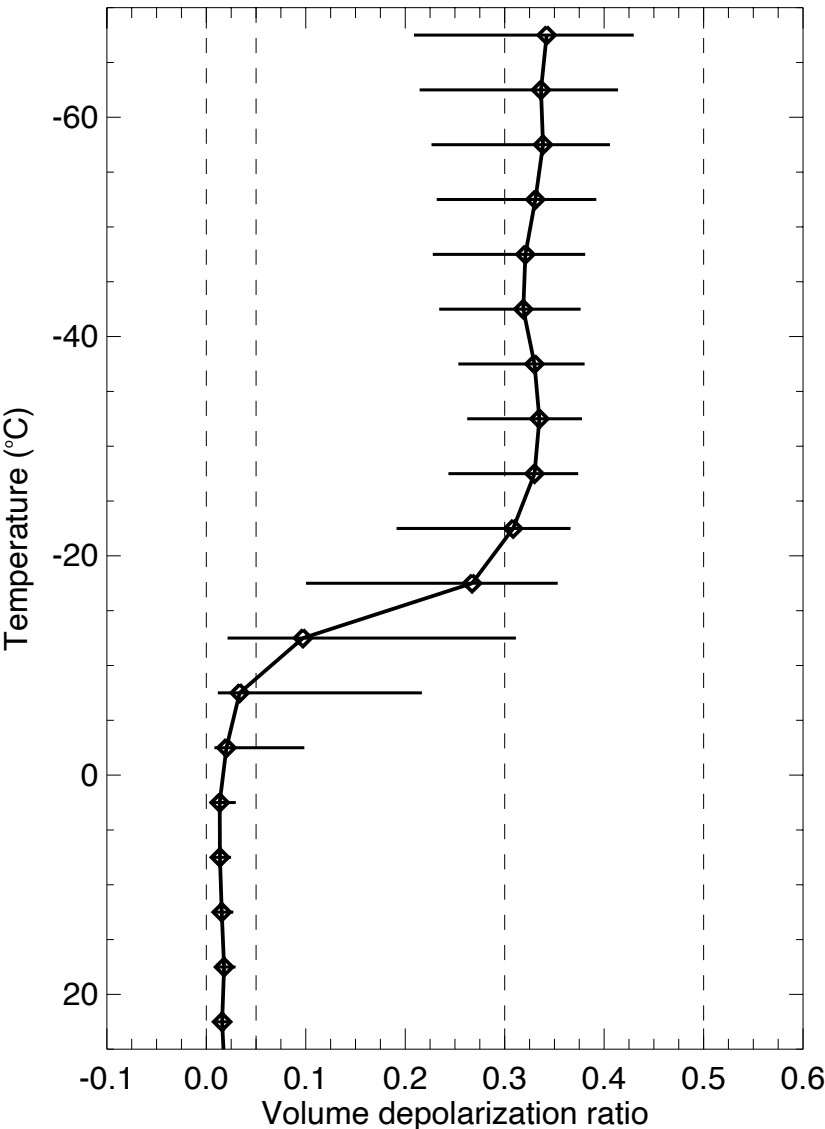

**Figure 4: Median volume depolarization ratio as a function of temperature for each altitude bin within all detected cloud layers at GSFC (2018 – 2019) in temperature increments of 5-C°. Horizontal bars indicate the interquartile range (IQR). Dashed vertical lines indicate the thresholds for the cloud phase diagnostics (CPD) as defined in Table 1. Temperatures above 25 °C are not displayed because of small sample and cloud precipitation.**






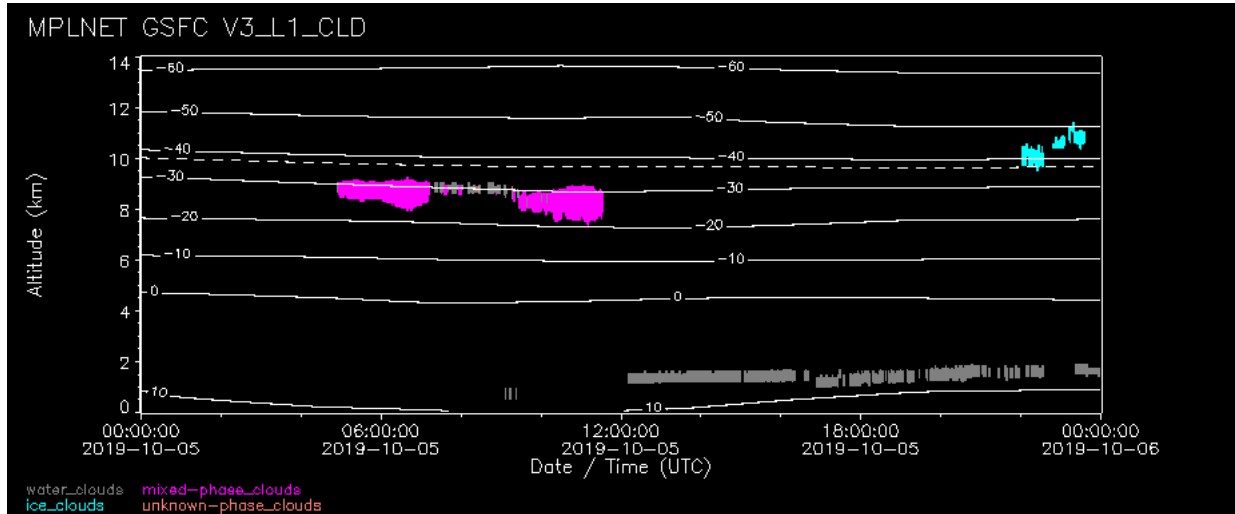

**Figure 5: Example of the cloud thermodynamic phase retrieval at GSFC on 5 Oct 2019. The phase mask indicates liquid water clouds (grey), mixed-phase clouds (magenta), ice clouds (cyan), and unknown phase (pink). The GEOS-5 temperature is shown by the contour lines (in 10-C° intervals). The -37 °C isotherm is indicated by the dashed contour line.**


![Figure 6: MPLNET GSFC V3_L1_NRB]

**Figure 6: Frontal cloud system at GSFC on 27 Mar 2018: NRB (top), volume depolarization ratio (middle) and phase mask (bottom). Altitude bins where the signal uncertainty is twice the signal strength have been suppressed for easier viewing. Note the**
**use of a log scale for the NRB. The phase mask indicates liquid water clouds (grey), mixed-phase clouds (magenta), ice clouds (cyan), and unknown phase (pink). The GEOS-5 temperature is shown by the contour lines (in 10-C° intervals). The -37 °C isotherm is indicated by the dashed contour line.**


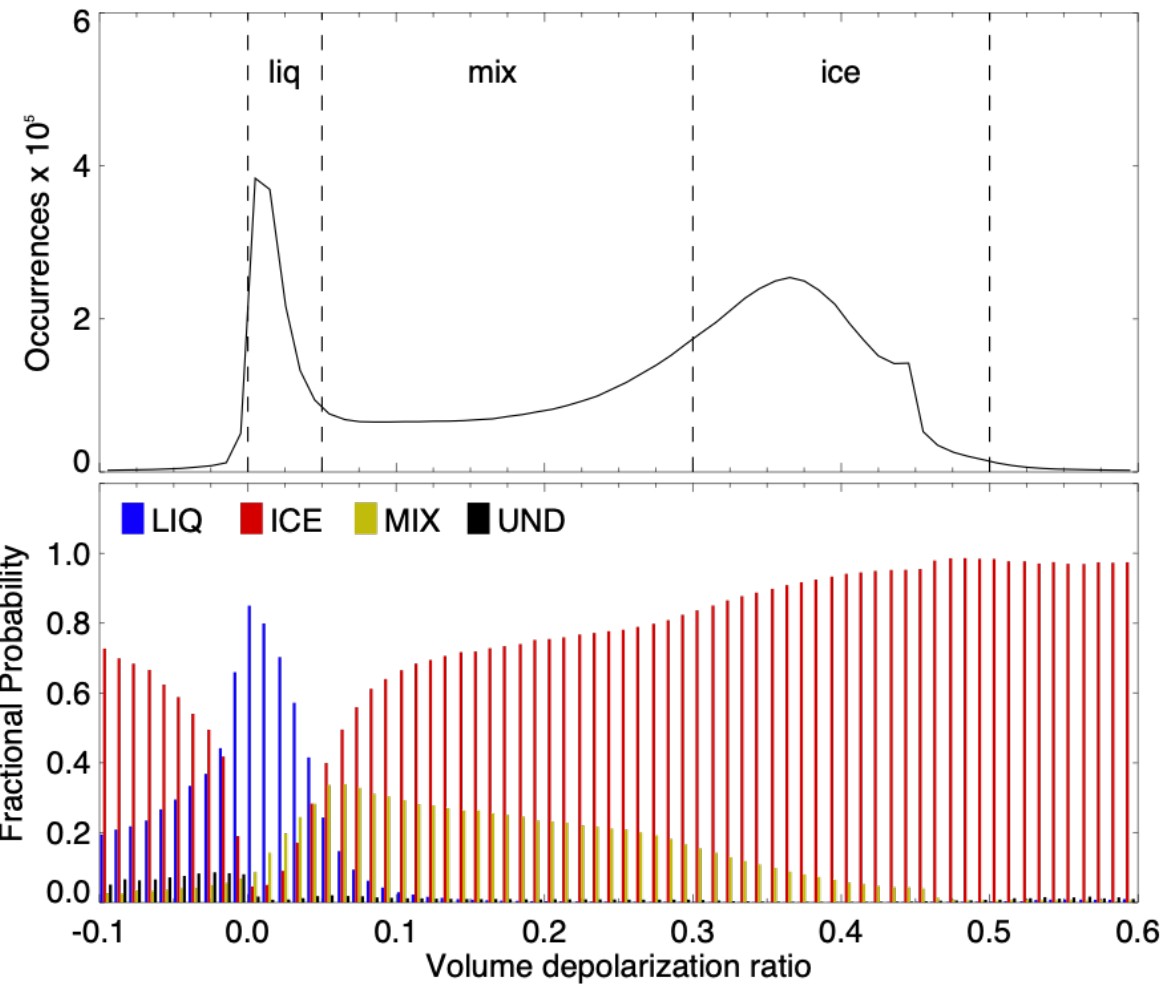

**Figure 7: (Top) Distribution of volume depolarization ratios for each altitude bin within all detected cloud layers at GSFC (2018 – 2019). The vertical dashed lines indicate the thresholds for the cloud phase diagnostics (CPD) as defined in Table 1. (Bottom) Fractional probability of retrieved cloud phases in 0.01 increments for the volume depolarization ratios shown in the top figure. Cloud phases written in lowercase letters of the top figure indicate these are altitude bin designations, while phases written in uppercase of the bottom figure represent the layer-determined designations.**




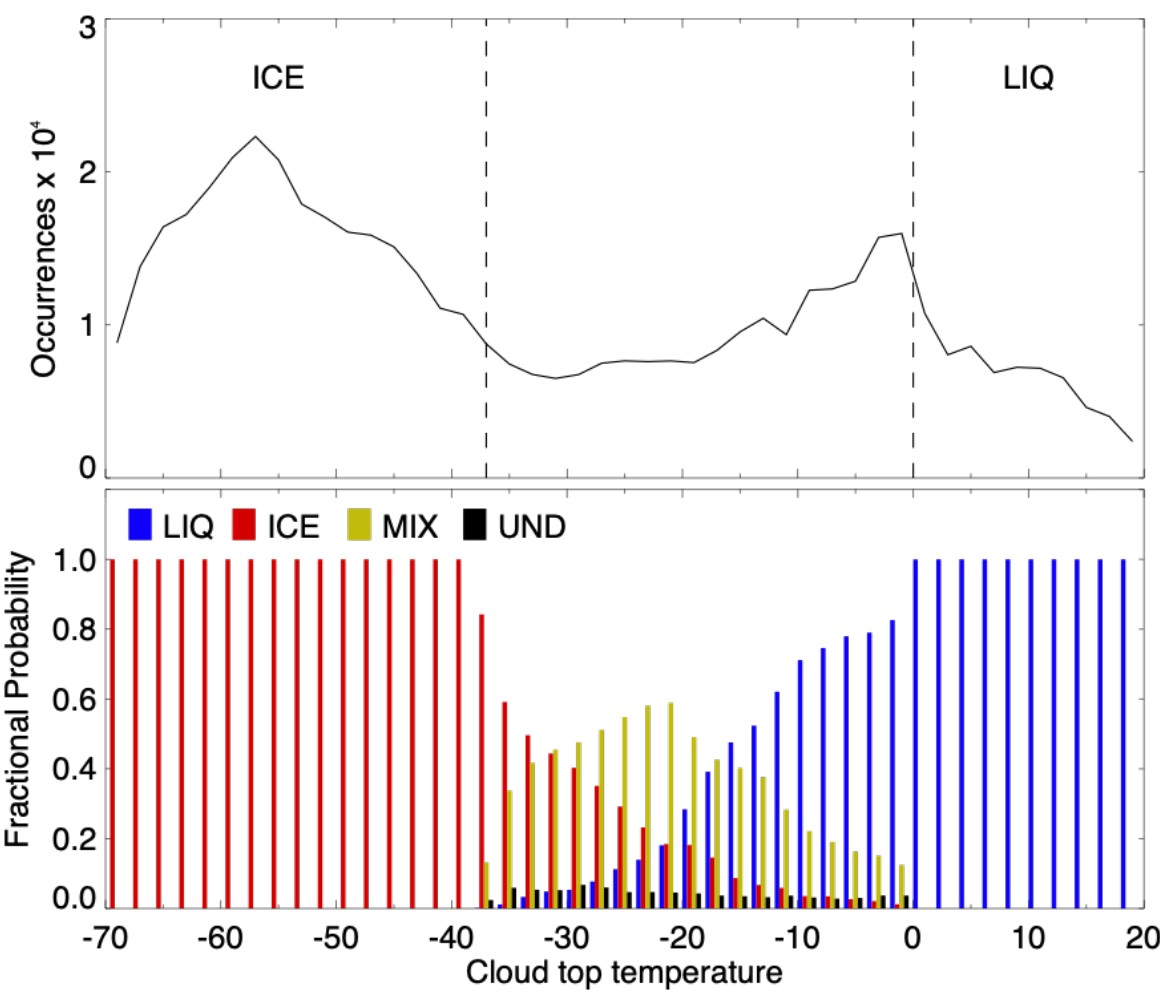

**Figure 8: (Top) Distribution of cloud top temperatures for all detected cloud layers at GSFC (2018 – 2019). The vertical dashed lines indicate the boundaries of the mixed-phase regime (-37 °C – 0 °C). (Bottom) Fractional probability of retrieved cloud phases in 2-C° increments for the cloud top temperatures shown in the top figure.**

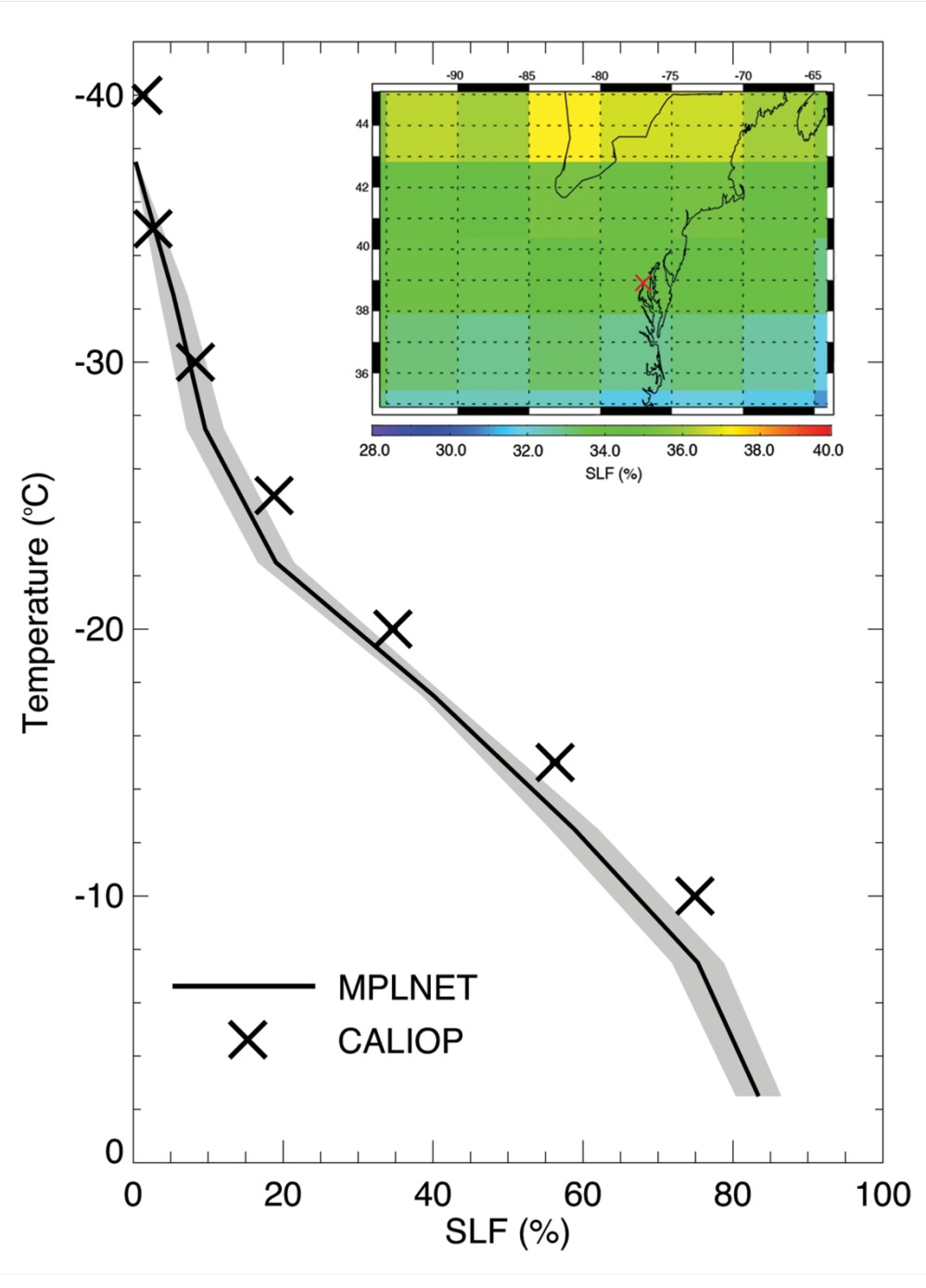

**Figure 9: Supercooled liquid fraction (SLF) averaged over GSFC (2015 – 2019) from MPLNET (solid line) and CALIOP (black ×) observations.** **The inset shows the horizontal distribution of CALIOP SLFs at the -20 °C isotherm surrounding GSFC (indicated by the red ×). The CALIOP SLF profile is calculated using the 2.5° latitude × 5° longitude grid box containing GSFC. The shaded area indicates the standard error for MPLNET observations. CALIOP standard errors are less than 0.7 at all isotherms.**
