# Peer review of "Determining Cloud Thermodynamic Phase from the Polarized Micro Pulse Lidar"

_Atmospheric Measurement Techniques, 2020_

## Referee Comment (RC1) · Anonymous Referee #1 · 12 Sep 2020

The manuscript presents approach for classification of ice, water and mixed-phase clouds basing on MPL polarization measurements. Authors are well known experts in the field of MPL development and data analysis and in this study they provide statistics for two years cloud phase retrievals. Due to low power of MPL, only volume depolarization ratio can be used, still authors demonstrate that significant amount of information about dependence of depolarization on the cloud temperature together with phase fraction probability can be derived. Paper is well and clearly written and scientific level of presentation is high. Paper can be published in AMT.

---

## Referee Comment (RC2) · Anonymous Referee #2 · 15 Sep 2020

This paper presents new measurements from a micro pulse lidar (MPL) with a depolarization channel and stationed at Goddard Space Flight Center, Greenbelt MD. The objective is to demonstrate that volume depolarization can be used to infer the water phase of clouds when used in conjunction with cloud top temperature from the GEOS-5 model. The paper is well written and presented with sound justification for its conclusions. I believe this paper should be published after addressing the minor points below.

I do not see the wavelength of this lidar. I think a table with the instrument specs would be useful.

Line 114 – 116. This sentence makes no sense: The cloud layer observed between 8 – 9 km CTT = -32.1 °C) exhibits NRB near the cloud top in both the co-polar and cross-polar signals compared to the signals nearer the cloud base.

[Figure]

Lines 120 - 124 Have the authors considered how signal attenuation may affect the depolarization measurement?

Figure 6. I cannot see any unknown phase clouds in this figure. The legend says they are pink, However, Magenta (used for mixed phase) is close to pink, so a different color should be used for unknown.

Figure 9. You show the CALIOP spatial distribution of SLF for the -20 C isotherm. It shows only a slight latitudinal dependence. Is that true for other temperatures? The area used for the comparison may be too large despite the spatially uniform data. When comparing a satellite measurement to a ground based measurement, you should not exceed ~2x2 degree. What happens to the comparison if you cut down considerably on the area used? Also it would be nice to have state boundaries on that map. It is a little hard to get one's bearings as it is.

[Figure]

---

## Author Comment (AC1) · 30 Oct 2020

We thank Anonymous Referee #1 for their careful reading of the manuscript and their kind words.

———————————————————————

---

## Author Comment (AC2) · 2 Nov 2020

We thank Anonymous Referee 2 for their careful reading of the manuscript and their helpful suggestions to improve this work. Below, we provide responses to specific comments by the reviewer.

*I do not see the wavelength of this lidar. I think a table with the instrument specs would be useful.*

The wavelength of the lidar (532-nm) is included in the following table which is added to the revised manuscript.

[Figure]

| Parameter | |
|---|---|
| Wavelength | 532 nm |
| Laser pulse energy | 6 – 8 $\mu$J |
| Repetition rate | 2500 Hz |
| Receiver diameter | 178 mm |
| Vertical resolution | 75 m |
| Temporal average | 60 s |

*Line 114 – 116. This sentence makes no sense: The cloud layer observed between 8 – 9 km CTT = -32.1 °C) exhibits NRB near the cloud top in both the co-polar and cross-polar signals compared to the signals nearer the cloud base.*

Thanks for catching this mistake. The word "higher" was missing from the original text and has been added in the revised manuscript:

The cloud layer observed between 8 – 9 km (CTT = -32.1 °C) exhibits higher NRB near the cloud top in both the co-polar and cross-polar signals compared to the signals nearer the cloud base.

*Lines 120 - 124 Have the authors considered how signal attenuation may affect the depolarization measurement?*

To the limit of attenuation, the depolarization measurement is reliable because the cross- and co-polar signals are affected equally. In this regime, multiple scattering effects (discussed in Section 2.2) are more of a concern. Once the signal becomes fully attenuated (also discussed in Section 2.2), depolarization measurements are no longer useful.

*Figure 6. I cannot see any unknown phase clouds in this figure. The legend says they are pink, However, Magenta (used for mixed phase) is close to pink, so a different color should be used for unknown.*

While admittedly the colors are similar, Figure 6 (as well as Figure 5) has no occurrences of unknown phase clouds. So, using a different color would not affect the im-
ages in these cases.

*Figure 9. You show the CALIOP spatial distribution of SLF for the -20 C isotherm.
It shows only a slight latitudinal dependence. Is that true for other temperatures?
The area used for the comparison may be too large despite the spatially uniform
data. When comparing a satellite measurement to a ground based measurement,
you should not exceed 2x2 degree. What happens to the comparison if you
cut down considerably on the area used? Also it would be nice to have state
boundaries on that map. It is a little hard to get one's bearings as it is.*

The slight latitudinal dependence at other temperatures is similar to that of the -20 °C
isotherm. For example, refer to Figures 2a, 3a, and 4a of Tan et al. 2014, which shows
the global SLF at -10 °C, -20 °C, and -30 °C, respectively.

*Tan, I., T. Storelvmo, and Y.-S. Choi (2014), Spaceborne lidar observations of the ice-
nucleating potential of dust, polluted dust and smoke aerosols in mixed-phase clouds,
J. Geophys. Res. Atmos., 119, 6653–6665,doi:10.1002/2013JD021333.*

The size of the grid box was selected in order to maximize sampling from the satellite
observations. A smaller grid box would not provide sufficient statistics for the CALIOP
supercooled liquid-water fractions. Furthermore, what we present here is not intended
as a coincident, direct cloud-to-cloud comparison. Instead, it is a comparison of inde-
pendent measurements of the same phenomenon within the same region. As such,
the size of the grid box is less important. We attempt to clarify our intent in the revised
manuscript as follows:

A comparison of SLFs derived from each instrument (CALIOP and MPLNET) averaged
from 2015 – 2019 is shown in Fig. 9. Instead of direct comparisons using coincident
overpass times of the GSFC site by the satellite, the comparison uses a statistical
approach to investigate the representativeness of the two independent datasets.

[Figure]

Finally, the state boundaries have been added to Figure 9.

**Fig. 1.**